# Self-Renewal Inhibition in Breast Cancer Stem Cells: Moonlight Role of PEDF in Breast Cancer

**DOI:** 10.3390/cancers15225422

**Published:** 2023-11-15

**Authors:** Carmen Gil-Gas, Marta Sánchez-Díez, Paloma Honrubia-Gómez, Jose Luis Sánchez-Sánchez, Carmen B. Alvarez-Simón, Sebastia Sabater, Francisco Sánchez-Sánchez, Carmen Ramírez-Castillejo

**Affiliations:** 1Centro Regional de Investigaciones Biomédicas, Universidad de Castilla-La Mancha, 02006 Albacete, Spain; mc.gilgas@edu.gva.es (C.G.-G.); paloma.honrubia@uclm.es (P.H.-G.);; 2HST Group, Department Biotechnology-BV, Centro de Tecnología Biomédica, Universidad Politécnica de Madrid, 28040 Madrid, Spain; marta.sanchez@ctb.upm.es; 3Oncology Unit, Hospital General de Almansa, 02640 Albacete, Spain; jluisss@sescam.jccm.es; 4Complejo Hospitalario Universitario de Albacete, 02006 Albacete, Spain; 5Laboratory of Medical Genetic, Faculty of Medicine, Instituto de Investigaciones en Discapacidades Neurológicas (IDINE), University of Castilla La-Mancha, 02006 Albacete, Spain; 6Oncology Group, Instituto de Investigación Sanitaria San Carlos, 28040 Madrid, Spain

**Keywords:** breast cancer, tumor initiating cells, PEDF, relapse, tumoral biomarkers, self-renewal

## Abstract

**Simple Summary:**

While more efficient cancer treatments have led to a better quality of life and to five-year survival rates of over 85% for localized tumors, these rates remain below 30% for metastatic tumors. Maintenance of these tumors is due to a population of resistant “cancer stem cells” characterized by their self-renewal. More specifically, the original malignant cell divides into a series of identical stem cells (with low division rate and high resistance to treatment) and other tumor cells that respond to treatment; in this way, when the original tumor disappears, the first cells form a new more resistant tumor and lead to metastasis. In an effort to force the tumor cells to divide into cells that are responsive to chemotherapy, leaving no “cancer stem cells” left to metastasize, we have studied the combination of the Carboxy-PEDF fragment with standard treatments. As our experiments show, this combination could be used to lower chemotherapy doses as well as reduce the likelihood of relapse and metastasis.

**Abstract:**

Breast cancer is the leading cause of death among females in developed countries. Although the implementation of screening tests and the development of new therapies have increased the probability of remission, relapse rates remain high. Numerous studies have indicated the connection between cancer-initiating cells and slow cellular cycle cells, identified by their capacity to retain long labeling (LT+). In this study, we perform new assays showing how stem cell self-renewal modulating proteins, such as PEDF, can modify the properties, percentage of biomarker-expressing cells, and carcinogenicity of cancer stem cells. The PEDF signaling pathway could be a useful tool for controlling cancer stem cells’ self-renewal and therefore control patient relapse, as PEDF enhances resistance in breast cancer patient cells’ in vitro culture. We have designed a peptide consisting of the C-terminal part of this protein, which acts by blocking endogenous PEDF in cell culture assays. We demonstrate that it is possible to interfere with the self-renewal capacity of cancer stem cells, induce anoikis in vivo, and reduce resistance against docetaxel treatment in cancer patient cells in in vitro culture. We have also demonstrated that this modified PEDF protein produces a significant decrease in the percentage of expressed cancer stem cell markers.

## 1. Introduction

The incidence of breast cancer has increased in recent years due to an aging population [1,2]. In fact, breast cancer is the leading cause of death among females in developed countries, although the implementation of screening tests and the development of anti-neoplastic therapies such as Trastuzumab has increased the probability of a cure in those patients [3,4]. However, relapse rates remain very high in different types of breast cancer [5,6], indicating a need for further study of new pharmacological drugs and diagnostic methods to stop relapse in patients. Relapse is mainly due to the tumor cell population’s resistance, characterized by their capacity for self-renewal, resistance to drugs, and a slow cellular cycle [7,8]. This last characteristic allows for the detection of this population since this produces long retained labeling [9,10,11]. The auto-renewal capacity of these cells is essential for stem cells to persist throughout the life of the organism.

The Pigment Epithelium-Derived Factor (PEDF) protein has been previously related to this self-renewal mechanism [12,13]. PEDF is a 50 kDa secreted glycoprotein that belongs to the serpin family group but does not share its inhibitory role [14]. The PEDF protein may induce cellular differentiation and promote apoptosis in a variety of tumor cells [10,11], and it is also able to inhibit tumor proliferation, vascularization, cell migration, and metastasis [15,16,17], affecting the division of fast tumor cells. In addition, PEDF has been described as a natural angiogenesis inhibitor [18] with neurotrophic [19] and immuno-modulatory properties [20], as well as being a regulator of tumor progression [21,22] and oxidative stress [23,24]. Moreover, PEDF is a niche-derived regulator of adult neural stem cells [12,25,26] that activates slowly dividing cells without inducing proliferation or differentiation [12,13]. In order to counteract this ability, supposing that tumor stem cells respond in the same way as normal stem cells, we propose the use of the carboxyl end of PEDF, previously shown to deplete neural stem cell niches in mice [12,25]. Therefore, the first approach in this study was to use PEDF and its carboxyl terminus to test whether there was the same effect on the described tumor stem cell population in both patient-derived tumor cells and stable tumor cell lines [27,28,29].

The mechanisms underlying most of PEDF’s functions are not completely clear. However, it appears that PEDF acts via multiple high-affinity ligands and cell receptors [20]. Studies by the Tombran–Tink group showed that secretion is necessary for PEDF activity and that PEDF could even migrate to the nucleus [30]. As has been widely described, PEDF is a pleiotropic molecule, with four receptors characterized to date [31]. Moreover, it presents two domains with clearly differentiated functions: an anti-angiogenic part and a second domain with neurotrophic properties, each activating different signaling pathways [32,33,34]. This fragmentation of the molecule allows us to take advantage of the different effects of its distinct domains on various signaling pathways, as described by other authors [35,36]. The pleiotropy of this molecule has been studied in different research groups; for example, Volper and collaborators described two epitopes of the N-terminal part of PEDF, designated peptides 34 mer (AA 44–77) and 44 mer (AA 78–121), which showed significant antitumor effects when conditionally expressed in human prostate cancer cells [37]. Another example comes from Becerra’s group [14,38], which created additional C-terminal expression truncations demonstrating that the N-terminal domain is responsible for this trophism. Finally, a thorough review of the different functions of PEDF related to the different domains and peptides derived from the molecule, their varying phosphorylation statuses, glycosylation, spatio-temporal regulation [39], and interaction with the different possible receptors can be found in the review by Kawaguchi et al. [40].

In our case, we consider the inhibition produced by the carboxy-terminal fragment (Cter-PEDF with amino acid residues 195–418) [41] on the crucial self-renewal capacity of the TIC population particularly interesting, as it may prevent tumor recurrence (seen in other tumor cell types), as we will attempt to demonstrate throughout the article. More specifically, we will use the modified carboxyl end of the PEDF protein (CTE-PEDF), with a glutamic acid instead of serine amino acid in position 227 (Ser 227). This serine is the only phosphorylatable residue in this fragment, and its substitution with a negatively charged amino acid allows us to study the effects in a more stable and repetitive manner, as dephosphorylation will not occur [42,43]. This is why we suggest a new therapeutic mechanism that consists of the co-administration of our modified carboxy-terminal PEDF protein fragment and chemotherapy.

Together with the fact that tumor-initiating cells (TICs) are capable of self-renewal, this indicates that this protein-signaling pathway could be important not only as an anti-neoplastic agent but also as a regulator of self-renewal in TICs and patient relapse. In order to detect TICs, four different epitopes were analyzed; these epitopes are implicated in the cells’ different processes and have been previously related to cancer stem cells in the literature. BCRP1 is a drug transporter from the ABC transporter family, which, while not as ubiquitous as other family members (such as MDR1), is commonly expressed in the stem cell population [44,45]. EpCAM is a transmembrane glycoprotein that is involved in cell signaling, migration, proliferation, and differentiation [46,47]. This whole process is closely related to the epithelial–mesenchymal transition, essential in the metastatic mechanism in which TICs could play an important role. CD133 is a pentaspan membrane glycoprotein that has been used as a stem cell biomarker since its discovery in 1999, although its function is still unknown. AC133 is a glycosylated-isoform of CD133, recognized by a specific antibody that has been described as a biomarker for human hematopoietic stem cells and different cancer stem cell models [48,49,50,51].

All these proteins have been involved in different metastatic processes and could be a target for new therapies. More interestingly, they can be indicators of the process’ progress. Moreover, they could shed further light on the mechanisms and the signaling pathways of these cells involved in the resistance to chemotherapeutic treatments. One of the cellular mechanisms that may be involved in these resistance processes is anoikis [52,53,54,55,56]. Anoikis is a form of apoptosis that occurs in anchorage-dependent cells and in a niche context when the cells detach from the surrounding extracellular matrix (ECM) and lose the connection to the surrounding nurse cells [56,57]; this is one of the processes we observed after the combination of chemotherapy and treatment with the carboxy-terminal part of the PEDF protein in our models. We have performed the assays in three different breast cancer cell lines: MDA-MB-231 and MCF7 cell lines and Pa00 patient-derived cells from a patient luminal B subtype tumor, estrogen receptor-positive, progesterone-receptor negative, and HER2-negative, with a high Ki67. Details are described in detail in the Materials and Methods.

Particularly in the case of breast tumors, different groups report the implication of PEDF in the progression of this type of tumor. Proteomic analysis of exosomes isolated from human malignant pleural effusions, including some breast tumors, detected the presence of previously unreported molecules such as SNX25, BTG1, thrombospondin 2, and the PEDF protein [58]. There are also articles presenting contradictory results regarding the decrease in PEDF linked to tumor progression in breast cancer [59], especially in hormone-dependent tumors [60,61], although some of them have suffered retractions. Some of these contradictory or difficult-to-explain results may be due to the spatio-temporal regulation of PEDF, different degrees of phosphorylation of its three serines, or its binding to up to four different receptors, allowing for this multiplicity of functions [39,62,63]. Many publications highlight that the biological functions of PEDF are diverse and multidimensional [58,64,65]. Some assays link the PEDF molecule to mechanisms that suppress breast cancer metastasis by regulating the epithelial–mesenchymal transition [66,67,68,69]; others report surprising observations, such as that PEDF constitutively activates p42/44 MAPK/Erk but paradoxically does not affect mitogenic signaling [70]. The angiogenic potential of PEDF as an antineoplastic therapy is particularly noteworthy [18,71,72,73,74].

In this case, we have addressed the problem of locating breast tumor-initiating cells and the study of their self-renewal mechanism by inhibiting its functionality and potential tumor recurrence. We also postulate that the PEDF signaling pathway could be a potential therapeutic target for the control of cancer-initiating cells’ self-renewal and tumor relapse, as has been postulated in other neoplastic models [43].

## 2. Results

### 2.1. Long-Term Label-Retaining Cells Exhibit Characteristics of Cancer Stem Cells in Cancer Cell Lines and in Patient Cells

MDA-MB-231 and MCF7 cell lines exhibit a cell population characterized by long-retained labeling growing either in adherent or mammosphere conditions (Figure 1A,B). A population of 0.9 ± 0.3% and 3 ± 1% of long-term label-retaining cells is present when the MDA-MB-231 cell line is grown in an adherent monolayer and in mammosphere conditions, respectively (Figure 1A). This result is also seen for the MCF7 cell line (Figure 1B), where the percentage of long-term label-retaining cells is 3.32% and 6.28% in monolayer culture and mammosphere conditions, respectively.

Ascitic fluid cells from patients with metastatic adenocarcinoma also present long-term label-retaining cells after 8 days in culture (Pa00 cells, Figure 1C). A methyl purple assay proves that LT− cells show the same growth rate as our control cells but have a higher growth rate than LT+ cells (Figure 1C). After a dose–response assay with docetaxel, LT+ cells are more resistant to chemotherapy than control or LT− cells (Figure 1D). The IC50 value is double in the case of LT+ cells compared to control or LT− cells (vertical lines in Figure 1D). Although the IC50 dose change is less than 1 nM, we thought it was meaningful to reflect this. Docetaxel treatment eradicates 53 ± 5% of LT− cells at 2 nM, while only 27 ± 3% of LT+ were affected by the same concentration. At high doses of chemotherapy, the growth curve returns to control levels. However, these are higher than the doses applicable in patients due to their undesirable side effects.

Finally, Pa00 LT+ and sorted LT− cells were injected in nude mice to study the carcinogenicity of those populations in vivo. This experiment shows that the tumor volume is similar when injecting LT− cells and control non-separated cells but smaller when injecting LT+ cells (Figure 1E). In addition, Pa00 LT+ cells formed tumors more frequently than LT− cells, even in very low cell density injections (Figure 1F).

In short, LT+ cells show a lower growth rate, and they are more resistant to chemotherapy than LT− or control non-separated cells. In addition, in vivo assays reveal a decrease in the frequency of tumor formation and an increase in tumor size in LT− cells when compared to LT+.

### 2.2. PEDF Modulate CSC Properties, Producing an Increase in Drug Resistance and Proportion of LT+ Cells 

PEDF chronic treatment produces morphological changes in the size and cytoplasm shape of Pa00 cells in culture (Figure 2A). We registered an increment of 170 µm^2^ at the cytoplasmic area in PEDF-treated cells (730 ± 80 µm^2^ in PEDF-treated cells and 560 ± 30 µm^2^ in the control medium); what it could be indicating is alterations in the cytoskeleton related to the tumorigenicity of these cells. However, there are no significant differences between the nucleus size with and without treatment (Figure 2B).

We have also studied the growing pattern of patient-derived cells with chronic PEDF treatment. This slower growth rate of treated cells has been quantified as 37 ± 1% fewer cells in treated compared to untreated PEDF cell cultures (Figure 2C). This effect translates into a lower tumor volume when the cells are injected in nude mice (Figure 2D), showing close to a 40% decrease in tumor xenograft volume from treated cells compared to controls (Figure 2D). The histological analysis showed that PEDF-treated tumors present smaller necrotic areas than control xenografts. Looking at the micrographs (Figure 2E) for the tumors from untreated cells, we can see large acellular areas (arrows in Figure 2E) compatible with acellular necrotic spaces. In contrast, the micrographs for tumors from PEDF-treated cells do not show these same areas, as the nuclei and cytoplasm of the cells are better preserved. PEDF treatment also produces compact growth of the tumoral cells, which present dense cytoplasms and more compact external matrices compared to the control tumors, which show acellular areas with cell detritus (Figure 2E). In the micrographs with immunofluorescence staining for BCRP1 and CD133 markers (Figure 2H–K), we again observe a lower number of nuclei in the controls than in the PEDF treatments, with larger acellular areas concordant with areas of necrosis and cell death as in panoramic Figure 2E. In these controls, there is a greater abundance of cells in different mitotic phases, indicated by white arrows (Figure 2H,J). These cells were also checked in a dose-response assay, and, despite their low growth rate, PEDF-treated cells exhibited higher IC50 values and resistant populations, as 47 ± 8% of PEDF-treated cells survived, which means there were 16% more resistant cells than in the control, where just 31 ± 2% of the cells survived at the maximum dose tested. This result implies higher drug resistance to docetaxel as compared to control cells (Figure 2F). We carried out the next assay to prove that the slow growth observed in vitro and in vivo is due to the appearance of a higher number of slow-cycle cells after PEDF treatment. The quantification of LT+ cells after 3DIV (days in vitro) revealed a significant increase after PEDF treatment compared to the control (Figure 2G), with nearly a three-fold increase in cell number. Examples in Figure 2H–K with insert details (squares of Figure 2H–K) in Figure 2L–O.

To sum up, PEDF-treated cells show changes in cytoplasm size, decreased cell cycle kinetics, increased drug resistance, and the capacity to produce tumors with not only a lower growth rate but also lower tumor volume and fewer necrotic areas.

### 2.3. Cter-PEDF Counteracts the Effects of Native PEDF, Decreasing the Resistance in Tumors and Inducing Anoikis and Depleting CSC

As a truncated protein, CTE-PEDF does not exhibit the same effects in stem cells as the full-size PEDF protein, so we decided to check if this peptide would also counteract the effects of the native protein in our cancer model. This protein presents phosphorylation sites that are important for the protein’s function. In order to maintain the negative charge, glutamate was introduced in position 227 of the protein’s sequence, where serine 227 was previously located. This modification leads to the CTE-PEDF protein (Figure 3A). We applied CTE-PEDF (200 ng/mL) in vitro treatment to patient-cultured cells, Pa00. This experiment shows an anoikis effect (Figure 3B) and morphological changes that can be measured by cytoplasmic and nuclear areas and cell distances (Figure 3C). Nuclear and cytoplasmic areas are reduced by 29% (nuclear area: control 110 ± 2 µm^2^, CTE 81 ± 2.34 µm^2^; cytoplasmic area control: 556 ± 18 µm^2^, CTE: 394 ± 17 µm^2^), while cell distances double in size (control 12 ± 1 µm, CTE: 23.0 ± 0.2 µm). Finally, treated cells divide faster but survive for less time than untreated cells in culture (Figure 3D).

The next approach was a dose–response assay with docetaxel, as previously performed with native PEDF treatments. The result demonstrates that CTE-PEDF-treated cells are less resistant (squares in Figure 3E) than control cells (circles in Figure 3E). Only the CTE-PEDF treatment produces a 20% reduction in the initial population. The IC50 value is double in the control group when compared to CTE-PEDF-treated cells (vertical lines in Figure 3E).

The results demonstrate that CTE-PEDF produces anoikis in cancer patient cells in culture and reduces resistance to docetaxel.

### 2.4. CTE-PEDF and Cter-PEDF Depletes Percentage of CSC Expressing Markers

To consider whether the decrease in resistance and anoikis is produced because of a reduction in CSC number and CSC expression markers, Pa00 cells were injected with Cter-PEDF or PBS acute treatment in nude mice. The resulting tumors were dissected and dissociated to study stemness marker expression via flow cytometry (Figure 4A). Four different epitopes, previously related to cancer stem cells, were analyzed: BCRP1, EpCam, CD133, and AC133. Cter-PEDF treatment produces a significant reduction in all studied markers in patient tumor cells Pa00 (Figure 4B). The same experiment was performed with cells chronically treated for three weeks in culture with CTE-PEDF. CTE-PEDF has a negative charge in glutamic 227 and is more stable than Cter-PEDF, as the latter has a phosphorylatable serine 227 residue, which can lose the negative charge when phosphorylated. The same reduction in marker expression was observed (Figure 4C).

In vivo, assays with xenografts (Figure 5A) of PEDF and CTE-PEDF chronically treated cells also show the influence of these proteins in the cancer stem cell population and resistant population. Immunofluorescence assays (Figure 5B) show differences in the expression of tumor stem cell-related markers when comparing the population of cells treated with PEDF to those treated with CTE-PEDF. A total of 3 × 106 cells were injected, with tumors appearing one week after the injection in every case. In both cases, there is a modification of the expression pattern of the BCRP1 and CD133 proteins. When exposed to chronic PEDF treatment, there is an increase in cells positive for these markers, while the opposite is observed after chronic treatment with CTE (Figure 5B). When treated with CTE, fewer cells are positive for BCRP1 and CD133 compared to controls or cells treated with PEDF. Figure 5C shows the relationship between the effect of PEDF and CTE on tumor cells, with the IC50 of docetaxel altered in both cases. PEDF treatment produces more resistant cells than the control, whereas CTE treatment decreases the IC50 of docetaxel, which coincides with the hypothesis that this treatment increases sensitization to the drug due to a higher division rate that thus prevents tumor cell survival.

Cells treated with PEDF, in addition to higher expression of markers, also show a higher concentration of LT+ cells (Figure 6A,B). These LT+ cells confer more chemoresistance to the culture (Figure 6C). We therefore set out to combat the effect of PEDF in vivo using the CTE peptide, which reduces marker expression and chemoresistance, as we have already stated.

To correlate the reduction in cancer stem cell marker expression in Pa00 cells (Figure 4) with decreased tumorigenicity (function), xenograft tests were carried out by injecting BCRP1-positive cells previously treated with CTE. The results show that PEDF increases LT+ cells (Figure 6A,B); these LT+ form more resistant tumors (Figure 6C), while the PEDF-treated cells take longer to form tumors, as shown in Figure 6D. Positive cells were injected at different concentrations, with their respective positive control of CTE-PEDF peptide treatment. Despite the heterogeneity of the in vivo experiments, it can be seen in Figure 6D that in most cases, the tumors from treated cells appear later than the untreated tumors. Even when injecting cells that had been previously treated with CTE, after injection, the xenograft appears earlier than in control cells. Figure 6D shows the assays performed on cells exposed to different successive radiotherapeutic treatments. Looking at expression assays by qPCR (Figure 6E), it is clear that the expression of p21, responsible for stopping the cell cycle in tumor stem cells, decreases. To test the effect of our model in these assays, we have performed new experiments with the CTE peptide and also with its homolog CTA (without phosphorylable serine, as here it is switched to an alanine). These assays mimic the effects seen with CTE combined with chemotherapy, as these cells also become less resistant to radiotherapy (contrary to what is seen with the negative control CTA) (Figure 6F). This could possibly be due to their faster division, causing those cells with a cell cycle regulated by p21 expression to disappear. Another cell type, such as Pa00, derived from patients with advanced breast cancer, also corroborates these results, as can be seen in Figure 5C. However, tumors formed from cells submitted to chronic CTE treatment are the least resistant of all to chemotherapy, as we have shown previously in Figure 5C. The effect of CTE is also synergistic with radiotherapeutic treatments, as it decreases cancer stem cell viability significantly with CTE treatment and radiotherapy (gray bars) as compared to radiotherapy alone or with a negative control peptide (CTA peptide without negative charge, white bars).

In summary, CTE-PEDF decreases resistance and putative stemness markers involved in self-renewal and patient relapse, demonstrating the interest in these proteins for potential applications controlling drug-resistant cell populations in patient samples.

## 3. Discussion

Breast cancer is the most common cancer among women worldwide. Today’s cancer treatments are increasingly effective and have fewer side effects, and our patients, especially breast cancer patients, have a better quality of life and a higher survival rate than a few years ago. However, we see that the global incidence continues to rise, with almost two million new cases per year due to an aging population and lifestyle. While the death rate from breast cancer remains stable in women under 50 (generally hereditary tumors), the death rate in women over 50 is decreasing by more than 1% per year, which also makes breast cancer the most prevalent tumor type. The five-year survival rate is over 85% for localized and regional tumors but less than 30% for those that metastasize. It is, therefore, advanced metastatic disease, maintained by the resistant cancer stem cell population, that should be of concern. We study how to prevent relapse, taking breast cancer as an example since its impact on society reaches an incidence of 15.53 (95% CI = 14.94 to 16.14) per 1000 PY [75]. The tumor subtype influences the risk of recurrence, ranging from 13% to 41%, depending on the tumor and the state of the nodes of the primary tumor [76,77] as hormone receptor-positive, metastatic breast cancer, and triple-negative breast cancer. It is important to distinguish tumor-initiating cells responsible for tumor formation and tumor relapse. No good spectra of specific markers are available for cancer stem cell detection. The expression of membrane determinants is a gradient during the development and maturation of biological processes, and even when testing a combination of the different epitopes found in the literature [78,79,80], it is difficult to find a completely specific cancer stem cell marker. However, tumor-initiating cells (TICs) are considered responsible for the slow cell cycle and quiescent characteristics of the tumor. Our primary goal was to demonstrate that TICs are important for tumor maintenance, resistance, and relapse. The second objective was to discover a possible new treatment to delay patient relapse. We demonstrate that PEDF (and derivatives) have a moonlighting role in breast cancer development and relapse. This has a potential application in clinical cancer therapies utilizing co-administration with chemotherapy to improve relapsed cancer treatment. This is the case for CTE-PEDF, which decreases drug resistance [42,43,81]. The co-administration with chemotherapy would lead to a less resistant population because TICs would have lost their stemness characteristics, as we discuss in the following paragraphs. The regulation of breast cancer progression through regulation of breast cancer stem cell-like properties has been recently described, probably mediated by the NF-κB cell signaling pathway [82,83], and the connection between chemoresistance, mesenchymal plasticity, and cancer stem cells in metastasis origin [84,85,86,87,88].

We have shown that MDA-MB-231 and MCF7 breast adenocarcinoma cell lines exhibit a cell population characterized by long-term retained labeling growing either in adherent or mammosphere conditions, as has been previously described in other tumor cell lines [27,28,29] and even in Pa00 cells [89]. Other researchers have detected similar percentages of stemness and invasiveness of breast cancer cells in these cell lines, which were modified by estrogen through Gli1 activation [90,91]. Even in mammosphere cultures, we have observed that a high DDAO-positive population, with slow cell cycle cells, remains after 8 days in culture; these cells have probably only divided once in this period. This subpopulation could be related to resistant cells maintained by IL-6 in some breast cancer treatments [92,93]. Also, they could be affected by the inhibitor of the human epidermal growth factor receptor 2 (HER2), lapatinib [94], and are our focus in this work.

Ascitic fluid cells from a patient with metastatic adenocarcinoma (Pa00 cells) have been used in this paper parallel to commercial cancer cell lines. Ascitic fluid, usually a result of inflammation events, contains malignant cells in up to 97% of patients with a neoplastic diagnosis and is a gold standard for pericardial carcinomatosis detection [94]. Ascitic fluid contains tumor cells that grow in culture conditions [95,96]. Part of these cells also present long-retained labeling (DDAO) after 8 days in culture, as we have demonstrated previously [89]. We selected these cells by flow cytometry sorting according to their high DDAO content, deemed LT+ cells. This experiment showed that LT+ cells grow slower and respond less to chemotherapy than non-separated control cells or LT− cells [92,97,98]. We postulated that this is the reason for the resistance observed in some treatments. Similar differences in resistance have also been observed in other colon and breast cancer cell lines [9,43] and even in other more aggressive types of tumors such as a glioblastoma C6 cell line [42,99]. In our hands, the final percentage of resistant cells after maximum physiological doses of docetaxel treatment is also higher in LT+-positive cells than in the rest. This LT+ population that supports higher docetaxel IC50 than LT− or control cells could be the origin of patient relapse. These same LT+ cells could also be related to higher BCRP1 and CD133 marker expression. To investigate this hypothesis, we injected Pa00 LT+ and sorted LT− cells in nude mice to study the tumorigenicity of those populations. We have considered not only the final volume of the xenograft tumors but also the timing of the tumors’ appearance. LT− cells formed tumors at the same time as control non-separated cells, but LT+ Pa00 took longer to form tumors. Accordingly, the percentage of final tumor growth after LT+ cell injection is 1/3 higher than in LT− xenografts. This result, together with the previously shown data, leads us to consider that the difference between these two cell types is the initial tumor capacity. These data correlate with previous data from other research groups, which connect chemoresistance, tumorigenicity potential, and slow cycling in some tumor cells [28,29,100]. Several other markers (such as TROP2 or KLF5) could be involved in this chemoresistance process, too, as has been recently identified by our group [101,102].

All these data support the hypothesis that LT+ cells present slow cell cycle division, higher chemotherapy resistance, and a higher frequency of tumor formation than LT− cells. These characteristics confirm the idea that LT+ cells could be involved in relapse and metastasis progression in breast cancer.

In our hands, PEDF chronic treatment produces a decrease in the growth rate in vitro and in vivo due to an increase in the number of LT+ cells (slow-cycling cells). The data shown in Figure 2 indicate that cells treated with PEDF have more controlled proliferation, with slower self-renewal processes than simple tumor cell division that ends in internal necrosis of the tumors.

As a conclusion from the cytoplasm measurements in Figure 2A,B, and the significant differences observed, we hypothesize that these PEDF-treated cells with a higher cytoplasm-to-nucleus ratio slow down their cell cycle. Several proteins have been related to this phenomenon. For example, the histone chaperone nucleoplasmin (Npm2) has been identified as a putative nuclear size effector [103] that binds histones and may play a key role in tumor development and progression [104].

Other cytoskeletal proteins have been related to this activity and cytoplasm size (cytomegaly). For example, non-muscle myosin IIA (MYH9) is a ubiquitously expressed cytoplasmic myosin that regulates the actin cytoskeleton, cell migration, cell polarization, and signal transduction in cancer cells. MYH9+ cells, size-differentiated by flow cytometry, showed strong CSC characteristics, including in vivo tumorigenicity, migration, invasion, cisplatin resistance, and positivity for CD133+ [105].

The data recorded by other investigators between cytoplasmic size and proteins involved in tumorigenicity will be tested in our system in the near future. For the moment, they coincide with the increased cytoplasm/nucleus ratio and the slower growth observed in our experiments. Cytoplasm size can be related to multiple physiological phenomena, among them quiescence and cell cycle exit. Although it is difficult to conclude what the increased cytoplasmic area means in our experiments, the data are in agreement with the hypothesis put forward and also with tumorigenicity and proliferation data observed by other authors, as argued.

The experiment that supports this optimal cellular state as a result of PEDF treatment is shown in Figure 2F, where it is seen that these cells are more resistant to chemotherapeutic pharmacological stresses and retain vital dyes for longer due to their slower cell division, as indicated in Figure 2G. Despite their low growth rate, PEDF-treated cells show higher IC50 values and resistant populations than untreated control cells, indicating higher resistance to chemotherapy. One possible explanation for this effect could be that the low growth rate lends more time for these cells to repair chemotherapy damage [106,107]. However, another phenomenon could contribute to this stage, such as higher expression of ABC drug transporters in these cells [108,109]. On the other hand, the effect that we attribute to the increase in slowly dividing cells could be due to other effects, such as the inhibition of angiogenesis, widely attributed and recently countersigned for the PEDF protein [110]. In addition to this anti-angiogenic effect, PEDF has been reported to have an impact on lymphangiogenesis [111]. Release of PEDF into the tumor microenvironment, among other factors, inhibits vascular growth while also promoting lymphangiogenesis associated with cancer [112]. Although PEDF has been recognized as an anti-metastatic factor, its role remains controversial due to some conflicting reports indicating a role in metastatic progression in some cases [113], diversifying PEDF’s impact on tumorigenicity, as our results point out. Current studies of the tumor microenvironment (TME), in which PEDF could be involved, offer an overview of the primary functions of each component of the TME in cancer initiation, progression, and invasion [112,114,115]. In MCF7 breast cancer cells, silencing of PEDF has been shown to promote resistance to tamoxifen [60]. It could be hypothesized that the PEDF signaling pathway is involved in the binding of the estrogen receptor to its activators and its proliferation activator function [21,63,116,117]. Some peptides derived from PEDF [118] activate the proliferation of progenitor or stem cells, as in our case. Whether our C-terminal peptide is involved in the regulation of resistance to tamoxifen will need to be tested in future experiments; this hypothesis would offer new potential uses for that peptide.

In contrast, we present here that PEDF treatment increases resistance to docetaxel, in agreement with what has been described about this molecule and the change in cellular metabolism in resistant breast cancer tumor cells [64,117]. PEDF is self-regulated by VEGF and the angiogenesis process. HIF1alpha-independent angiogenesis could be under interference from both PEDF’s anti-angiogenic effect and tumoral cells’ resistance [119]. In culture, however, the anti-angiogenic effect of PEDF could be shielded by its self-renewal effect over cancer stem cells. These results could also be explained based on the metabolic pathway in which cytochrome P450 metabolizes both components (PEDF and docetaxel). Blocking of PEDF by fluvoxamine, a P450 inhibitor, has been previously described [120]. P450 polymorphisms present in each of the cell lines, and patient cells analyzed could account for these differences.

Returning to the possible effectiveness of the CTE-PEDF peptide, treated patient cells showed symptoms compatible with anoikis and morphological changes in culture. The anoikis effect is the apoptosis induced by lost, insufficient, or inappropriate interactions between the cell and the extracellular matrix [121,122]. Some of the cell-signaling pathways involved in cancer progression (such as BMPs) have also been related to proliferation, anoikis resistance, metastatic migration, and drug resistance of breast cancer cells [87,88,102,123]. This anoikis effect could be the basis for the interaction of TICs with the surrounding niche and could be involved in the loss of stemness properties and chemotherapy resistance [98]. Docetaxel resistance decreases when cells are treated with CTE-PEDF. It counteracts the effect of PEDF, which is a niche protein [12] involved in self-renewal and trophic maintenance of pluripotent cells. PEDF is secreted by endothelial cells and is one of the proteins involved in the self-renewal capacity of stem cells. Endothelial cells also play a significant role in tumor progression and metastasis [124,125], and the niche signals could be key to understanding tumor progression, epithelial–mesenchymal transition, and distant metastasis at patient relapse. A significant difference is shown between treated cells and controls, with PEDF or derived peptides, but the protocol followed in this experiment should be standardized to obtain a better result. The molecular subtype used in this work could be influencing the observed effects. However, the same results have been obtained in cell lines of other types of tumors [42,43]. Also, we observed differences in the level of expression of proteins that participate in the PEDF signaling pathway, justifying the differences between cell lines in the different experiments while maintaining the final effect and the extractable conclusions.

In short, the experiments carried out indicate that the carboxyl terminal fragment of the PEDF protein, CTE-PEDF, could be an effective chemo-sensitizing agent since it is capable of counteracting the effect of the endogenous native PEDF protein, as had been previously demonstrated in neural stem cell populations [12]. Expression of markers that had been previously related to cancer stem cells also decreases when cells are treated with CTE-PEDF. This effect confirms the idea that CTE-PEDF produces a decrease in resistance to drugs such as docetaxel because it reduces the number of tumor-initiating cells. This result indicates a possible potent application as a future treatment for this protein when co-administered with chemotherapy [111,113]. What we postulate here is that the modification of the PEDF protein—that is, the C-terminal peptide can affect the cancer root through a mechanism of competition with native PEDF. Previous experiments have shown this competition and that increasing doses of truncated peptide can inhibit the effect of the native protein [12]. Although further studies need to be carried out, these experiments point to a possible new strategy for the treatment of cancer relapse.

In order to verify the importance of our data, we have compared our results with other kinds of treatments. In addition to chemotherapeutic treatments, radiotherapeutic adjuvant treatment is also widely used in breast cancer [126,127]. We have shown that radiotherapeutic treatment decreases cell cycle markers related to stem cell self-renewal, such as the cell cycle protein p21 [128,129], which decreases with progressive radiotherapy treatment. All combined, the results of this comparison are of great applicability in resistant tumors, especially in cases of triple-negative breast cancer, as these show the worst response to current treatments. There are many experiments to be performed in this regard, and these will be future lines of research in the laboratory. However, the data obtained so far seems to be of importance, contributing to general knowledge and permitting us to advance in the fight against resistance to cancer treatment.

To conclude, our hypothesis can be summed up in the schematic below (Figure 7). There has been much discussion in the literature regarding the idea of a countdown mechanism that limits cellular replicative potential by giving it an intrinsic heuristic value [130]. We consider that cancer stem cell depletion is of particular importance, and CTE can help us in this direction. In the summary figure, we can see how PEDF enhances the self-renewal of tumor stem cells, akin to what occurs in normal stem cells [12,25]. Molecular markers involved in the cell cycle of cancer stem cells, such as p21, increase their expression after PEDF treatment, thereby indicating an increase in the self-renewal of these cells, as well as their perpetuation and, as a consequence, the chemoresistance of these cell cultures. In contrast, chronic treatment with CTE-PEDF allows the extenuation of the tumor stem cell population, along with an increase in the proliferative rate, but also better response to chemotherapy treatments, since active cycle cells are more susceptible to apoptosis from anti-neoplastic treatments, thus decreasing the harmful rate of resistant cells (Figure 7).

## 4. Materials and Methods

### 4.1. Cell Culture

MCF7 (ATCC^®^ HTB-22™), MDA-MB-231 (ATCC^®^ HTB-26™), and 293 HEK-293 (ATCC^®^ CRL-1573™) cell lines were acquired from ATCC (Manassas, VA, USA).

MCF7 are epithelial cells isolated from the breast tissue of a 69-year-old white, female patient with metastatic adenocarcinoma with Pleural effusion. The MCF7 line retains several characteristics of differentiated mammary epithelium, including the ability to process estradiol via cytoplasmic estrogen receptors and the capability of forming domes. The cells express the WNT7B oncogene [94].

The MDA-MB-231 cell line was isolated at M D Anderson from a pleural effusion of a patient with invasive ductal carcinoma and is commonly used to model late-stage breast cancer [120]. This cell line is ER, PR, and E-cadherin negative and expresses mutated p53. In microarray profiling, the MDA-MB-231 cell genome clusters with the basal subtype of breast cancer. Since the cells also lack the growth factor receptor HER2, they represent a good model of triple-negative breast cancer. MDA-MB-231 cells are invasive in vitro and, when implanted orthotopically, produce xenografts that spontaneously metastasize to lymph nodes [120].

The Pa00 cell line was derived in the laboratory from a patient luminal B, estrogen receptor-positive and progesterone receptor-negative, HER2-negative, with a high Ki67. She received chemotherapy treatment with adjuvant CMF (classical cyclophosphamide, methotrexate, and 5-fluorouracil treatment) and tamoxifen. The patient suffered a bone recurrence, receiving multiple lines of hormonal and chemotherapeutic treatment, including exametasane, megefren, faslodex, taxotere, cisplatin, liposomal doxorubicin (caelyx), vinorelvin, capecitabine, and spinal radiotherapy. Subsequently, she suffered visceral progression with ovarian and hepatic involvement. There were also carcinomatous ascites, of which a sample was collected for in vitro culture [89], in which we found a small population of cells that express stem cell cancer markers, as can be seen in the results of the present article.

Pa00, MCF7, and MDA-MB-231 cells were maintained in DMEM (Lonza, Basel, Switzerland) 10% FBS (Lonza), 1% glutamine (Lonza) (0.2 M), and 1% penicillin/streptomycin (Lonza) (100 units + 100 µg/10 µL) in a 5% CO_2_ humidified incubator at 37 °C.

### 4.2. Staining with DDAO and Sorting

Cells for cell cycle dynamic assays were plated (100,000 cells). Next, these cells were washed with PBS, disaggregated, and then incubated at 37 °C with Cell Trace ^®^ Far-Red-DDAO-SE (DDAO-SE, Molecular probe ref C34564, Eugene, OR, USA) at the concentration recommended in the product data sheet. After 10 min, an aliquot of freshly labeled cells (approx. 200,000 cells) was fixed with 0.5% of paraformaldehyde to use as a positive control for the experiment. Then, we centrifuged the cells at 180× *g* for five minutes, and we suspended them in the culture medium to grow under standard conditions. Eight days in vitro later, CM-spf cells were disaggregated and sorted via In-fluxTM (Becton Dickinson, Franklin Lakes, NJ, USA) sorting equipment depending on their fluorescent retaining labeling level. Positive cells were considered slowly dividing cells and thus potential tumor-initiating cells.

### 4.3. Cytometry Assay

Cytometry assays were performed with a MACSQuant Analyzer 10 cytometer (Miltenyi Biotec ref 130-096-343, Bergisch Gladbach, Germany). Samples were first washed in PBS and incubated with Miltenyi Biotec FcR blocking Reagent (human) in 100 microliters of sample. The immunostaining was performed per the standard protocol recommended by the commercial houses of the different antibodies. The antibodies and the working dilutions used were as follows: AntiBCRP-FITC 5D3 Chemicon (Temecula, CA, USA) and AntiBCRP-PE 5D3 Chemicon incubated for 20 min at a 1:10 dilution, while AntiEpCAM- PE Clone HEA-125, AntiAC133-PE Clone AC133, AntiCD133-PE 293C3, Anti CD44-FITC and CD24-PE, all from Miltenyi Biotec, were incubated 10 min at a 1:10 dilution. Sorting experiments were performed with a BD Influx™ cell sorter (Franklin Lakes, NJ, USA). FITC or PE are used as an abbreviation of fluorescein isothiocyanate and phycoerythrin, respectively.

### 4.4. Gene Expression

For gene-expression assays, total RNA from cultured cells was extracted using the RNeasy Mini kit (Qiagen, Hilden, Germany) and used immediately for reverse transcriptase reactions or stored at −80 °C until use. RNA concentration was measured in a Nanodrop spectrophotometer (Thermo Fisher, Waltham, MA, USA). cDNA synthesis was performed using the RevertAid First Strand cDNA Synthesis kit (Fermentas, Waltham, MA, USA). The reaction was prepared according to the manufacturer’s instructions. Quantitative PCR reaction was performed for CDKN1A (also known as p21 inhibitor) using the KiCqStart SYBR Green kit (Sigma, St. Louis, MO, USA) according to the manufacturer’s instructions. KiCqStart SYBR Green predesigned primers (Sigma, St. Louis, MO, USA) were employed (H_CDKN1A_1: Forward primer 5′ TGTAAAACGACGGCCAGT and reverse primer 5′ CAGGAAACAGCTATGACC). Primers were used in a final concentration of 300 nM, and 10 ng of cDNA was used per well for a total volume of 10 µL. All cDNA samples were measured in triplicate in a 96-well plate covered with adhesive seals in the thermocycler Roche LightCycler 480 (Roche, Basel, Switzerland). Reactions started with 10 min at 95 °C, followed by 45 cycles of 15 s at 95 °C, 1 min at 60 °C, and 10 s at 72 °C. The 2^−ΔCT^ method was used for calculating the normalized mRNA expression. Beta-actin was used as a housekeeping gene to normalize samples. Fold increase relation over untreated controls has been used to graphically represent the decrease in gene expression.

### 4.5. Xenografts

FOXn1nu females, at 1 month of age, were obtained from Charles River International Laboratories (Wilmington, MA, USA) to use in these experiments. Animals were housed and bred under 20–25 °C, 50–60% humidity, and 12 h light–dark cycles. All experiments were performed in accordance with relevant guidelines and regulations, and the animals were treated in accordance with the approval of the local ethics committee (University of Castilla-La Mancha PI081746). The experiments were performed as previously described [30]. In short, untreated cells (control) and treated cells (8 nM CTE-PEDF) suspended in PBS were injected subcutaneously into both flanks of immunocompromised mice. A total of 5000 cells were injected in a final volume of 200 µL of a 1:1 dilution with MatrigelTM Basement Membrane Matrix (Becton Dickinson) with a 25-gauge needle. Tumor growth was monitored weekly with a caliper. The final tumor volume was calculated as V = 2 × L1 × L2 × π/6. Tumors were mechanically and chemically dissociated with collagenase and trypsin at 37 °C. They were then washed with PBS and seeded in DMEM medium, 10% FBS, 1% glutamine (200 mM), 1% penicillin/streptomycin, 0.5% EGF, and 0.04% FGF in a 5% CO_2_ humidified incubator at 37 °C 12 h before the start of other experiments.

### 4.6. PEDF and Cter-PEDF, CTE-PEDF Production

PEDF and the carboxy-terminal part of the PEDF protein (Cter-PEDF) were cloned into pcDNA™3.1/myc-His A, B, & C Mammalian Expression Vectors (InvitrogenTM, Carlsbad, CA, USA) following the protocol previously described by Sánchez-Sánchez and coworkers [41]. After cloning, vectors were checked by sequencing (3130 Applied Biosystems. Foster City, CA, USA). Serine 227 was mutated into a glutamate (E) residue using ser227glu-dw (5′-CCA AGT AGA AAT CCT CGA GCT CAG TCT TTC TGG AGT-3′) and ser227glu-up (5-GTT TGA CTC CAG AAA GAC TGA GCT CGA GGA TTT CTA-3) primers. After mutagenesis, the Cter-PEDF peptide with this glutamic change was named CTE-PEDF. Proteins were produced by HEK-293-T cells after transfecting with phosphate calcium. Conditioned mediums were collected after 3 DIV, quantified via Western blot using anti-c-myc mouse monoclonal IgG1 (Santa Cruz Biotechnology, Dallas, TX, USA) and anti-phosphoserine clone 4A4 (Millipore, Burlington, MA, USA) and stored at −20 °C or purified using GE Healthcare Life Sciences™ HisTrap™ FF Crude columns (Thermo Fisher Scientific, Waltham, MA, USA).

### 4.7. Treatment with PEDF and CTE-PEDF

Cells were treated with PEDF or CTE-PEDF at a final concentration of 8 nM in the medium. The acute treatment consists of a single treatment: two hours of peptide exposure before chemotherapy treatment. Chronic treatment consists of six peptide treatments (culture medium with CTE once per week), totaling six weeks of peptide exposure before chemotherapy treatment.

### 4.8. Treatment with Radiotherapy and CTE-PEDF

Cells were plated in a 24-well plate at a final concentration of 15,000 cells/well in a total volume of 200 microliters. Half of the plate was plated with CTE-PEDF medium. The next day, radiotherapy to 2 Gy, 4 Gy, and 6 Gy was applied via a Siemens PRIMUS X6MV LINAC. Field size 10 × 10 cm^2^, depth 10 cm, SSD = 100 cm, Siemens PRIMUS X6MV.

### 4.9. Dose–Response Curve

Cells were plated in a 24-well plate at a final concentration of 15,000 cells/well in a total volume of 200 microliters or 200 cells/well in a 96-well plate for sorted cells. Half of the plate was plated with CTE-PEDF medium. The next day, the drug was added in decreasing concentrations: 4 nM, 2 nM, 1 nM, 0.5 nM, and 0.25 nM. These were cultured for 3 days under drug exposure in in vitro conditions and then revealed via MTT assay or methyl purple assay.

### 4.10. MTT Assay

The cell culture media was removed, after which 100 microliters/well of 3-(4,5-dimethylthiazol-2-yl)-2,5-diphenyltetrazolium bromide (MTT) was added. After 30 min of incubation, the supernatant was removed, and the precipitated crystals were dissolved in 100 microliters of DMSO. The plate was read in a spectrophotometer at 540 nm.

### 4.11. Methyl Purple Assay

This method [43] is used to quantify surviving cells. A total of 5000 cells/well were seeded in 24-well plates, with a final volume of 250 µL/well. The next day, cells were treated with increasing doses of chemotherapeutic agents and stored in a humidified incubator at 37 °C, 5% CO_2_ for 4 days. Then, cells were fixed with 0.5% glutaraldehyde (Sigma, St. Louis, MO, USA) for 10 min. Next, cells were stained with 0.1% crystal violet for 20 min. After several washes with PBS, 10% acetic acid was used to solubilize the sample. Finally, a spectrophotometric reading was performed at a wavelength of 590 nm. The IC50 value was determined as the dose necessary to eliminate 50% of the cell population, obtained through logarithmic regressions made with the DE.0 plus v 1.0 program.

### 4.12. Histology

Cryostat sectioning slides were performed using a cryostat (Microm HM 550, Thermo Scientific). Tumor samples were fixed with formaldehyde at 4%, placed in sucrose at 30% overnight, included into Tissue-Tek^®^ OCT™ (Sakura ^®^ Finetek USA, Torrance, CA, USA), and then frozen with liquid nitrogen. Slides (12 µm) were stained with hematoxylin, mounted with Dako Ultramount Aqueous Permanent Mounting Medium (Agilent Technologies, Santa Clara, CA, USA), and analyzed with a Leica-DMRXA-photomicroscope (Leica, Wetzlar, Germany).

### 4.13. Analysis of Cell Morphology

A total of 500,000 treated and control cells were seeded in p100 plates. After 24 h, cells were incubated for 30 min with Hoechst (5 µg/mL). Ten random microscopy images were taken (Motic AE31, Barcelona, Spain) using phase contrast and ultraviolet light, depending on the experiment. Analysis of the cytoplasmic area and the separation between cells was analyzed using the Image J (version 1.51) freely available software (https://imagej.nih.gov/ij/ accessed on 11 November 2023). The images are binaries, and the cells are surrounded by a mask (example in Figure 2A) so that the program can quantify the areas of these masks. For nuclear area counting, the areas occupied by the nuclei masks of 10 cells per image are summed by averaging 10 random images and using a total of 100 cells per condition. For the area calculation, the areas occupied by the 10-cell masks per image are summed and averaged over 10 images, using 100 cells per condition (the same as in the nuclear area count), and the area of the nuclei calculation is subtracted. Therefore, 200 cells were quantified, 100 cells per condition.

### 4.14. Immnuno Assays

A total of 500,000 treated and control cells were seeded on p100 plates with glass coverslips on the bottom to allow for removal of the cells once grown. After chronic treatment (4 weeks) with PEDF, the cells were fixed with 1% paraformaldehyde and stained with specific BCRP1 (MAB4155A4, Sigma-Aldrich, St Louis, MO, USA) and CD133 (MAB4399-I, Sigma-Aldrich) antibodies according to the manufacturer’s instructions. At the end of immunolabeling, the cells were incubated for 30 min with Hoechst (5 µg/mL). Fluorescence microscopy images were taken with a Motic AE31 microscope equipped with an ultraviolet lamp.

### 4.15. Measurements of the Xenograft Tumors

From six days and three times a week after inoculation of the tumor, the area where the cells were inoculated was palpated to detect the presence of masses attributable to the injected tumor, and thus the latency time of tumor appearance in each animal could be established. Photographs were taken for qualitative monitoring of the tumors. When tumors exceeded three millimeters in diameter, the largest and smallest tumor length (mm) of each animal was measured twice a week using a caliper. To determine tumor volume, the following formula was applied: π/6 × major diameter × (minor diameter)^2^, with results expressed in mm^3^.

### 4.16. Statistical Analysis

Data were analyzed with R software 3.5.1 version (https://www.r-project.org/ accessed on 11 November 2023). The statistical analysis was carried out using Mann–Whitney U tests (one-tailed, significance level = 0.05). The data are expressed as the mean plus the standard error (SE). At least n = three independent experiments were performed for every assay. The results obtained are considered statistically significant when *p* < 0.05 (*), *p* < 0.01 (**), and *p* < 0.005 (***). Cell data were first transformed into a quadratic variable to improve data homogeneity. Then, normality was tested using the Sapiro–Wilk test, Q-Q plots, and Levene’s test for homogeneity of variance. Next, groups were compared using the non-parametrical Kruskall–Wallis test by ranks and Wilcoxon’s unpaired test to analyze data in pairs. Finally, logistic regression was used to model the relationship between the number of cells and progression. The odds ratio was used to strengthen the association between variables (confidence interval of 95%).

## 5. Conclusions

The PEDF stem cell self-renewal modulator protein modifies the carcinogenicity of cancer stem cells and could be a useful tool to control tumoral self-renewal and therefore patient relapse. We have designed a transgenic peptide derived from PEDF to interfere with the self-renewal capacity of cancer stem cells, inducing anoikis in vivo and reducing resistance in cells from cancer patients. We have also shown that this PEDF-derived peptide produces a significant decrease in the expression of cancer stem cell markers, making it a potential tool for delaying patient relapse.

## Figures and Tables

**Figure 1 cancers-15-05422-f001:**
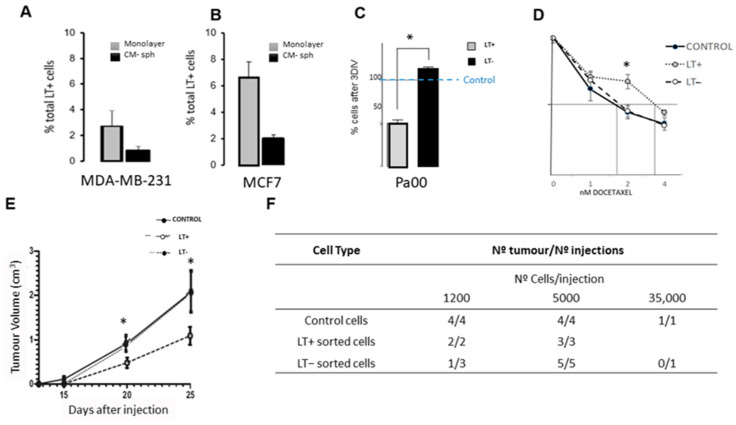
Long-term label-retaining cells (LT+) are present in cell lines and cultures from patient cells and display cancer stem cell characteristics. (**A**) MDA-MB-231 cells were stained with DDAO and cultivated 8DIV (days in vitro) in monolayers or as mammospheres (CM-sph). (**B**) Also, the MCF7 cell line showed an LT+ population, which was higher in mammosphere assays (cytometry assay) than in monolayers. (**C**) Growing patterns of LT+ and LT− cells from patient Pa00 cells stained and grown 8DIV (400 cells/well) and then sorted according to their DDAO content. The number of living cells after 3DIV was checked by methyl purple assay. LT− cells grew similar to controls and faster than LT+ cells. The growth of the control cells has been represented with a blue horizontal line. (**D**) Docetaxel dose–response curves for LT+, LT−, and control cells. Pa00 cells were stained with DDAO and grown for 8DIV, sorted by their content of DDAO, and grown with increasing concentrations of docetaxel. LT+ cells showed greater resistance against docetaxel than LT−, and respective IC50 are marked by perpendicular lines. (**E**) 5000 Pa00 cells were injected in nude mice in each case. The tumor volumes are similar when injecting LT− and control non-separated cells but smaller when injecting LT+ cells. All tumors were palpable at the same time. LT+ tumors grew slowly compared to those in the control or LT− group. (**F**) Table of number of injected mice and tumor formation of the different cell types (*n* = 3 all experiment; * *p* < 0.05).

**Figure 2 cancers-15-05422-f002:**
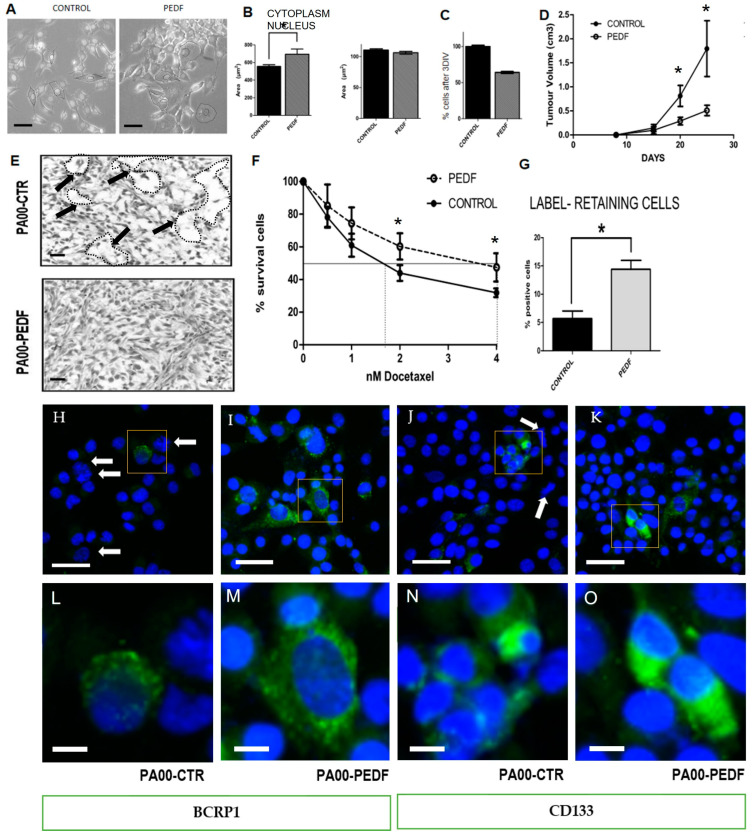
Pigmented Epithelium-Derived Factor (PEDF) increases the number of LT+ cells and the docetaxel resistance of breast cancer cells. (**A**) Cells treated with chronic PEDF showed a different morphology than control cells. (**B**) Quantification of morphological differences induced by PEDF treatment. Cells in the micrographs were measured via ImageJ program using the mask shown in Figure (**A**). (**C**) Growing pattern after 3DIV of PEDF treated cells and control. PEDF chronically treated cells grew slower than control. (**D**) Docetaxel dose–response curve of PEDF chronically treated cells and control. PEDF chronically treated cells were more resistant against docetaxel. (**E**) Histology of PEDF treated tumors and control tumors. Optical microscopic observation shows visible changes in cells chronically treated with PEDF, with a qualitative increase in both cytoplasmic density and the external matrix. Necrotic areas are bigger in controls compared to those with PEDF treatment. Black arrows show mask examples of areas compatible with acellular necrotic spaces. (**F**) Docetaxel dose–response curve of LT+ PEDF treated cells and LT+ untreated cells, *n* = 3. LT+ PEDF chronically treated cells were more resistant to docetaxel than untreated ones. (**G**) PEDF-treated cells grew more slowly than control cells, meaning that the dye-retaining population is up to three times larger than with PEDF treatment. (**H**) BCRP1 marker immunohistochemistry in control cells. (**I**) BCRP1 marker immunohistochemistry in PEDF-treated cells. (**J**) CD133 marker immunohistochemistry in control cells (**K**) CD133 marker immunohistochemistry in PEDF-treated cells. White arrows in (**H**–**K**) shown examples of the abundance of cells in different mitotic phases. (**L**–**O**) Insert at high magnification of the yellow square in (**H**–**K**). *n* = 3 in all experiments; * *p* < 0.05. Bar in (**H**–**K**) is 50 µm. Bar in (**L**–**O**) is 10 µm.

**Figure 3 cancers-15-05422-f003:**
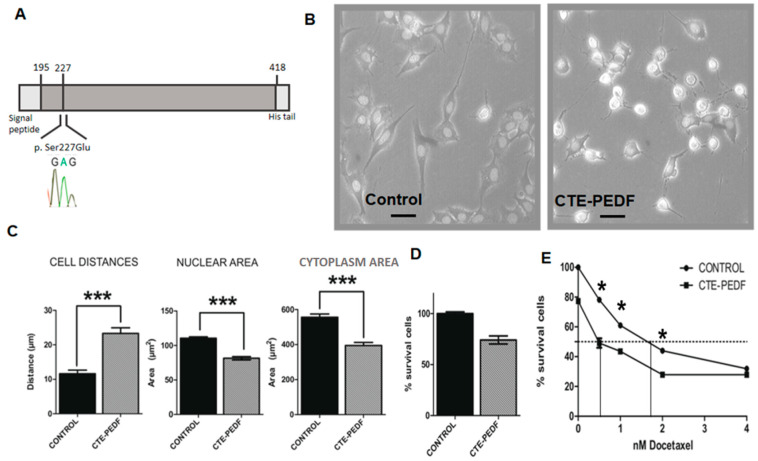
CTE-PEDF induces loss of anchorage and cell death in vivo and reduces resistance against docetaxel. (**A**) CTE-PEDF construction from 195 to 418 aa of PEDF and with glutamic acid instead of serine in position 227. (**B**) Pa00 cells were treated chronically with CTE (200 ng/µL). After a week, they showed an increase in anoikis figures. (**C**) Quantification of CTE-PEDF induced morphological changes in nuclear and cytoplasmic areas and cell distance. (**D**) Growing pattern after 3DIV of CTE-PEDF treated cells and control. Treated cells survive for less time than control cells. (**E**) Docetaxel dose–response curve of CTE-treated cells and control. CTE chronically treated cells were less resistant to docetaxel (*n* = 3 in all experiments; * *p* < 0.05, *** *p* < 0.001). Bar 50 µm.

**Figure 4 cancers-15-05422-f004:**
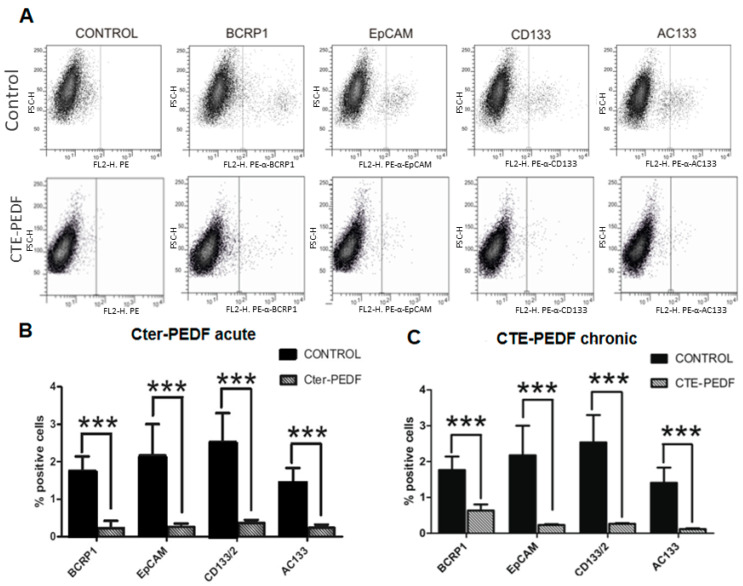
Cter-PEDF and CTE-PEDF treatments decrease in vivo cancer stem cell markers. (**A**) Cytometry assay showing the smaller number of CSC in treated cells compared to control. (**B**) Quantification of positive cells in acute treatment compared to control. (**C**) Quantification of positive cells in chronic treatment compared to control (*n* = 3 in all experiments; *** *p* < 0.001).

**Figure 5 cancers-15-05422-f005:**
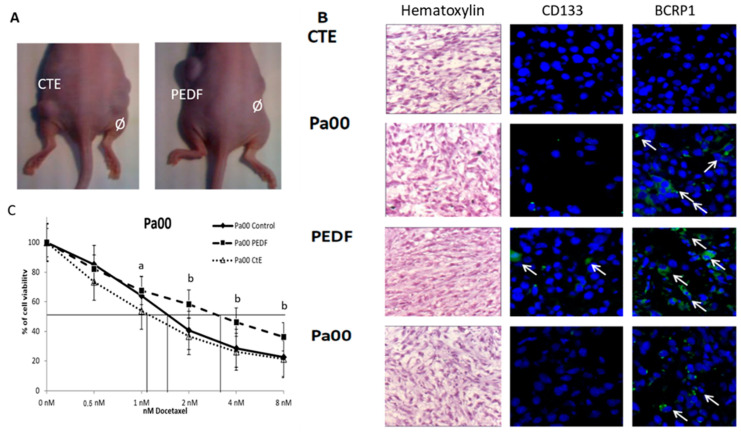
PEDF and CTE-PEDF treatments modified in vivo cancer stem cell markers. (**A**) Examples of xenografts of the PEDF-, CTE-, and control-injected cells. (**B**) Immunofluorescence of cells positive (arrows) for CD133 and BCRP1, respectively, in xenografts of chronic treatments. Immunocytochemistry with hematoxylin is shown at 20×. Immunocytochemistry assays with CD133 and BCRP1 are shown at 40× with DAPI nuclear staining. (**C**) Differential effects on cell viability of PEDF and CtE-PEDF treatments over Pa00 tumoral cells in culture. a: Statistically significant differences between Cter and control or PEDF-treated cells, *p* < 0.05. b: Statistically significant differences between PEDF and control or Cter-treated cells, *p* < 0.05.

**Figure 6 cancers-15-05422-f006:**
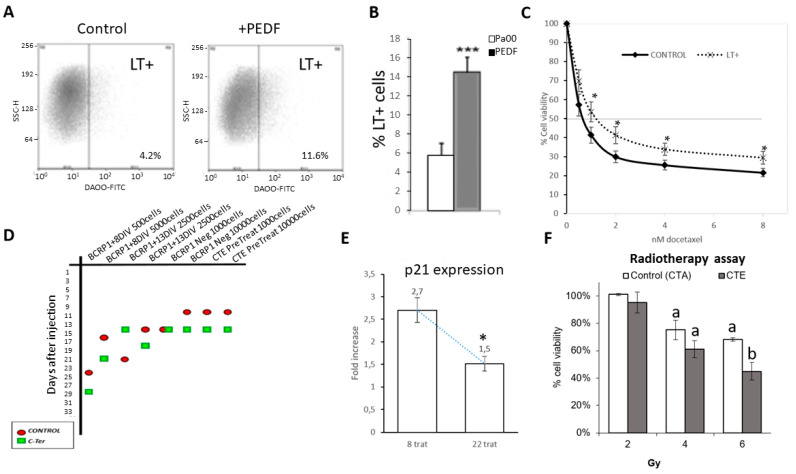
LT+ and BCRP1+ cells used in xenograft assays showing the effect of CTE-PEDF treatments in vivo and the synergy of this effect with chemo and radiotherapy. (**A**) Cell cytometry of PEDF-treated and untreated cells. LT+ cells are more abundant after PEDF treatment. (**B**) Quantification of three independent flow cytometry experiments in A. (**C**) Xenografts resulting from cells positive for stem cell markers and slow-cycling LT+ cells are more resistant tumors when subjected to dose–response assays after dissection and placed in cell culture. (**D**) Cells expressing the cancer stem cell marker BCRP1 were injected at different cell concentrations. Control-negative cells were also injected. In all cases, an assay was performed with and without treatment with the carboxyl end of PEDF (Cter-PEDF). The xenograft tumors of the Cter-treated cells appear time-delayed (in green) with respect to their controls (in red). (**E**) Consecutive treatment with radiotherapy (8 treatments and 22 treatments at 6 Gy) downregulates stem cell cancer markers such as the p21 mRNA involved in cell cycle arrest of these cells. (**F**) The effect of CTE is also synergistic with radiotherapeutic treatments, as it decreases cancer stem cell viability significantly more with CTE treatment and radiotherapy (gray bars) than with radiotherapy alone or with a negative control peptide (CTA peptide without negative charge, white bars). a: statistically significant differences (*p* < 0.05; compared to the non-irradiated control. b: statistically significant differences compared to 6 Gy treatment without CTE. (*p* < 0.05; *; *p* < 0.01; ***).

**Figure 7 cancers-15-05422-f007:**
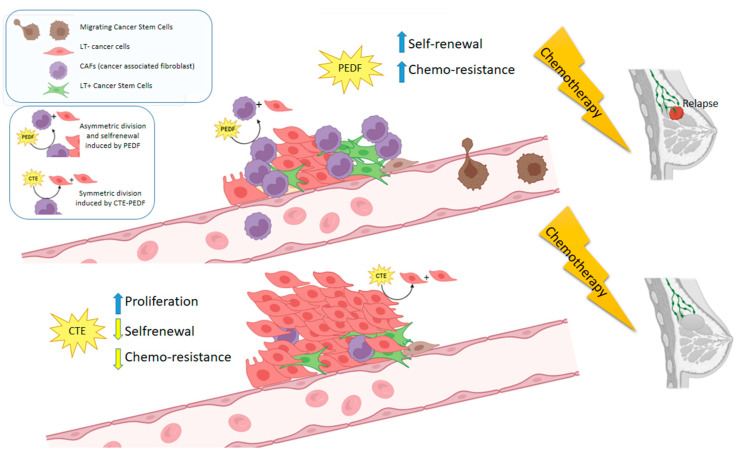
Hypothesis of the exhaustion and depletion of tumor stem cells by blocking the PEDF self-renewal factor signaling pathway. This could result in increased proliferation of chemotherapy-responsive cells and decreased tumor resistance and relapse frequency.

## Data Availability

Data are contained within the article.

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
