# Peer review of "Self-Renewal Inhibition in Breast Cancer Stem Cells: Moonlight Role of PEDF in Breast Cancer"

_cancers, 2023, doi:10.3390/cancers15225422_

Round 1

Reviewer 1 Report

Comments and Suggestions for Authors

 Carmen Ramírez-Castillejo has submitted a manuscript describing the role of PEDF in breast cancer. Overall, the manuscript and research were well conducted, however, I have the following points for authors to consider in their revised version. Furthermore, the manuscript must be thoroughly proofread and image quality must be improved.

Fig 2B and Fig 3C: How are the cytoplasm, nuclear area, and cell distances measured from DIC images? Explain the methodology in detail.

How many cells were quantified?

Fig.2E: explain how the cytoplasm density and external matrix are quantified from histological images represented in this figure.

Fig.2H and fig 5B: Quantify the number of BCRP1 cells and CD133 cells from the immunofluorescence data. A magnified image of the expression will be more convincing.

Fig.5A: Present the representative images of all the xenografts to show if there is any difference in tumor size.

Fig.7: the semiquantitative PCR is performed for n>3 assays as the authors have mentioned. Why not quantify the expression from three assays and plot them as a graph?

Minor:

Fig. 2B, Typo “CITOPLASM”

Fig. 5B Typo, it will be “Hematoxylin

Fig.6F : RADIOTHERAPY

Fig.6A: Put a high resolution image for the FACS plot, 6B and 6D-graph.

Fig8: It will be Tumor cells and not Tumoral cells. This model can be improved to make it a more visually appealing graphical abstract.

Methods section 4.3: It will be “CLONE”

4.2: It will be 0.5%

Comments on the Quality of English Language

No comments

Author Response

Dear Dr (Reviewer 1) Thank you for your helpful comments, as they have helped us improve the quality of the article and understanding of the data. I will now try to address each of your comments.
1. - Fig 2B and Fig 3C: How are the cytoplasm, nuclear area, and cell distances measured from DIC images? Explain the methodology in detail. How many cells were quantified?
We have included detailed explanations of the methodology used for analysis of the cytoplasm area and the separation between cells, in lines 227-228 of the text and material and methods. In summary, using ImageJ software, images are turned into binaries and the cells are surrounded by a mask (example in figure 2A), so that the program can quantify the areas of these masks. For nuclear area counting, the areas occupied by the nuclei masks of 10 cells per image are summed by averaging 10 random images and using a total of 100 cells per condition. For the area calculation, the areas occupied by the 10-cell masks per image are summed, averaged over 10 images, using 100 cells per condition (the same as in the nuclear area count) and the area of the nuclei calculation is subtracted. Therefore 200 cells were quantified, 100 cells per analyzed condition.
2. - Fig.2E: explain how the cytoplasm density and external matrix are quantified from histological images represented in this figure.
In the text we only attempt to reference the morphological description observed visually under the microscope. No quantified data are available and explicit reference to qualitative observations has been incorporated in lines 231 to 233.
3. - Fig.2H and fig 5B: Quantify the number of BCRP1 cells and CD133 cells from the immunofluorescence data. A magnified image of the expression will be more convincing.
BCRP1 and CD133 cells with and without treatment have been quantified using flow cytometry (Figure 4), which we consider the most appropriate technique for quantifying a large number of cells and evaluating minority populations, such as in this case. Electron micrographs and immunofluorescence were used for cell visualization and morphological corroboration of expression. However, we do not consider that the optical microscopy technique is suitable for this type of quantification, as it would require a very high number of fields to have an adequate representation and data with sufficient rigor and strength. This is considered a morphological and qualitative technique, in which quantification would not provide significant additional data compared to the other experiments that have already been carried out and detailed in the text. Neverthless to include understanding of the data we have added an insert at higher magnification for each of the images (2H – K).
4. - Fig.5A: Present the representative images of all the xenografts to show if there is any difference in tumor size.
Because the tumor cannot be seen in its entirety on the surface of the xenograft in many cases, thus requiring opening of the skin to show the size of the tumor, we consider xenograft images to be qualitative graphical documents. For this reason, we chose the two-diameter measurement and a conventional formula to quantify the differences in tumor size as can be seen in figure 2D and in previous experiments in figure 1E.
A section specifying the strategy and formula used to measure tumours has been added to the material as well as in the Methods section, lines 807- 815 of the document.
5. - Fig.7: the semiquantitative PCR is performed for n>3 assays as the authors have mentioned. Why not quantify the expression from three assays and plot them as a graph?
We have removed figure 7 as the quality of the membranes is not sufficient to quantify and represent the data in a meaningful way.
6. – Various minor concerns.
We have addressed the relevant typos and added higher resolution images where possible.

Reviewer 2 Report

Comments and Suggestions for Authors

In the manuscript titled “Self-renewal inhibition in breast cancer stem cells: moonlight 3 role of PEDF in breast cancer” the author Castillejo et al. presented interesting research on identification of Cancer stem cells (CSC) or Tumor initiating cells (TIC) in the Breast cancer cells. The author has identified the long -term label retaining cells as the CSC population in MDA-MB-231 and MCF7 cell lines, because of slow proliferation and quiescent nature. The author mentions that the slow proliferating and self-renewing LT+ cells are CSCs  and causes relapse and resistant to the chemotherapy regimens. Further, the author has tried to dissect out the mechanism of self-renewal and chemo-resistance in the cancer cells and found Pigment Epithelium- derived factor (PEDF) to be responsible for the same. The author has showed the PEDF treatment on Pa00 cells has increased the LT+ cell percentage compared to control. The PEDF treatment changes the cellular property and nuclear to cytoplasmic ratio. The observation was further supported by the increase in expression of BCRP1 and CD133 cell surface markers due to PEDF treatment, concluding PEDF is responsible for the stemness and chemoresistance in the TICs.

The authors further studied the role of PEDF by inhibiting its function and subsequently change in the cell proliferation. The inhibition of PEDF with carboxyl terminal domain of PEDF (Cter-PEDF and CTE-PEDF) resulted into changed nuclear to cytoplasmic ratio, increased proliferation of cells and improved response to the chemotherapy. At the end, the authors proposed a combination treatment regimen to avoid the probability of relapse by adding PEDF blockers and chemotherapeutics. I found the study compelling but lacks certain merits and incomplete results.

I have comments and concerns about the study and would like the authors to address these:

Major comments:

1.    I found there is incomplete information on the use of PEDF, how come the authors decided to use PEDF without any background analysis?

2.    The change in cells from flat to spherical maybe due to the cytotoxicity of CTE. I would like to know if the author has titrated the conc. of CTE to be used?

3.    Nuclear area of the LT+ cells show no difference compared to LT- cells, but how come the CTE-PEDF treated cells has lower nuclear area (Fig. 3)?

4.    I don’t see a drastic difference in the IC50 of LT+ cells upon PEDF treatment 0.5 nM difference can’t be represented as big change.

5.    I’m curious to see the FACS plot for the CD133, BCRP1 along with LT+ cells, in addition to p21 expression due to Radiotherapy. Radiotherapy is misspelled as Raditerapy.

6.    I found blocking PEDF for therapy contradictory as other reports suggests PEDF has anti-tumor role and significantly downregulated in breast cancers (Hong, H., Zhou, T., Fang, S. et al. Pigment epithelium-derived factor (PEDF) inhibits breast cancer metastasis by down-regulating fibronectin. Breast Cancer Res Treat 148, 61–72 (2014). https://doi.org/10.1007/s10549-014-3154-9). How the author is going to explain that?

Minor comments:

1.    I found the manuscript is not curated well in terms of data representation, there are so many mistakes in the text. In Fig. 1 (a,b) I would label Y-axis as % total LT+ cells and cell types either on top or bottom of graph. The graphs are not clear as there is inconsistency on the graph origin on X-axis.

2.    Fig. 1c, Y-axis percentage cannot be more than 100% percent.

3.    I would like to see the highest concentration of Docetaxel where survival of cell reduces to 10% or so.

4.    In Fig. 1, I found unconventional abbreviations such as MLY (monolayer), CM- sph/spf (Mammosphere). Inconsistency in mammosphere abbreviation.

5.    Drug resistance graph or IC50 plot should be represented in a better way.

6.    Fig. 2. Cytoplasm is misspelled as Citoplasm, Cytoplamatic etc.

7.    In Fig. 2 (H-K) besides ICC, I’m curious to know the percentage of population on FACS.

8.    Better FACS plot with population percentage (Fig. 6A).

Comments on the Quality of English Language

There is lot of incidences where the words are misspelled, sentences need to be rephrased.

Author Response

Dear Dr 

Thank you for your helpful comments, as they have helped us improve the quality of the article and understanding of the data. I will now answer each of your questions:

1.- How is it that the authors decided to use PEDF without any background analysis?

Three introductory paragraphs have been introduced in lines 44 to 50; 58 to 64, and 70 to 81, along with more relevant literature to explain the background of our hypothesis. We have also included a reference that details the different domains found in PEDF that regulate signaling and consequently affect the differential functions attributed to the protein; this can be found in table 1 of the Kawaguchi et al. article, as well as other similar articles, including the crystallographic characterization of the protein.

2.- The change in cells from flat to spherical maybe due to the cytotoxicity of CTE. I would like to know if the author has titrated the conc. of CTE to be used?

Indeed, as you have observed the cell morphology shown in Fig 3B coincides with that of a possible cytotoxic effect. This is what we describe and further detail in the results section, where we refer to possible anoikis, or cell damage due to loss of adhesion to the substrate. What is ruled out based on the literature is the possibility of direct cytotoxicity, since in vivo experiments show no cytotoxicity when CTE is injected via peristaltic pump into mouse brains (Nature Neuroscience 2006 doi: 10.1038/nn1657).

  1. Nuclear area of the LT+ cells show no difference compared to LT- cells, but how come the CTE-PEDF treated cells has lower nuclear area (Fig. 3)?

We do not fully understand the reviewer's conclusion that the nuclear area of LT+ cells shows no difference compared to LT- cells. If you refer to figure 2B, what is shown is a trend but no significant difference is found between the nuclear size of control and PEDF-treated cells.

In the case of cells treated with CTE-PEDF, these have a smaller nuclear area, possibly due to faster division cycles, hence their greater sensitivity to chemotherapy and lower capacity for self-renewal. Like all cells in a constant division cycle, these have characteristic features such as condensed chromatin (heterochromatin): thick, dark granules that indicate that the DNA is transcriptionally inactive in mitotic prophase, with nuclei staying compacted and condensed until the nuclear membrane disintegrates. These can be differentiated from the larger nuclei of cells in G0, which show dispersed chromatin (euchromatin) that occupies a larger volume when DNA is being transcribed.

  1. I do not see a drastic difference in the IC50 of LT+ cells after PEDF treatment. A difference of 0.5 nM cannot be represented as a large change.

We assume that the reviewer is referring to the changes observed in figure 1D. It is true that this is not a drastic difference, although the chemotherapy dose reaching the IC50 of LT+ cells is twice that of control cells or LT- cells, and that the IC50 of many tumor lines for this drug is around 0.5nM to 1nM (Kelland & Abel, 1992). However, to highlight this we have added the following sentence at line 184: "Although the IC50 dose change is less than 1nM, we thought it was meaningful to reflect this".

  1. I am curious to see the FACS plot of CD133, BCRP1 and LT+ cells, plus p21 expression due to radiotherapy. Radiotherapy is misspelled as Raditherapy.

We have corrected the typo of Radiotherapy. Unfortunately, we don't fully understand the question posed by the reviewer, as the plots of CD133 and BCRP1 are shown in figure 4. If this does not answer the question, please tell us what your query is so as to try and answer it properly, thank you.

  1. I found that blocking PEDF for contradictory therapy as other reports suggest PEDF has an antitumor role and significantly down-regulated in breast cancers (Hong, H., Zhou, T., Fang, S. et al. Pigment epithelium-derived factor (PEDF) inhibits breast cancer metastasis by down-regulating fibronectin. Breast Cancer Res Treat 148, 61-72 (2014). https://doi.org/10.1007/s10549-014-3154-9). How is the author going to explain that?

Thank you for the comment and the reference, which obviously needs to be cited and commented. We have inserted a paragraph to this effect in lines 142 to 150, and the relevant references in addition to the one you ask for:

“Some of these contradictory or difficult-to-explain results may make sense of the spatio-temporal regulation of PEDF, its different degree of phosphorylation on three serines along the molecule or its binding to up to four different receptors, allowing for this multiplicity of functions(Abooshahab et al., 2023; Filiz & Dass, 2012; Kawahara et al., 2020). Many publications highlight that the biological functions of PEDF are diverse and multidimensional(Fitzgerald et al., 2012; Hong et al., 2014; Subramanian et al., 2012). Some assays link the PEDF molecule to mechanisms that suppress breast cancer metastasis by regulating epithelial-mesenchymal transition(Belkacemi et al., 2018; Tsuruhisa et al., 2021; Zhou et al., 2016, 2017). And others report such surprising observations as that PEDF constitutively activates p42/44 MAPK/Erk, but paradoxically does not affect mitogenic signalling(Jones et al., 2023).”

Abooshahab, R., Hooshmand, K., Salami, H.-A., & Dass, C. R. (2023). The Impact of Pigment-Epithelium-Derived Factor on MCF-7 Cell Metabolism in the Context of Glycaemic Condition. Pharmaceutics, 15(8). https://doi.org/10.3390/pharmaceutics15082140

Belkacemi, L., Atkins, J. L., Yang, L., Gadgil, P., Sater, A. K., Chow, D. S., Bose, R. N., & Zhang, S. X. (2018). Phosphaplatin anti-tumor effect enhanced by liposomes partly via an up-regulation of PEDF in breast cancer. Anticancer Research, 38(2), 623–646. https://doi.org/10.21873/anticanres.12267

Filiz, G., & Dass, C. R. (2012). Reduction in tumour cell invasion by pigment epithelium-derived factor is mediated by membrane type-1 matrix metalloproteinase downregulation. Pharmazie, 67(12), 1010–1014. https://doi.org/10.1691/ph.2012.2067

Fitzgerald, D. P., Subramanian, P., Deshpande, M., Graves, C., Gordon, I., Qian, Y., Snitkovsky, Y., Liewehr, D. J., Steinberg, S. M., Paltán-Ortiz, J. D., Herman, M. M., Camphausen, K., Palmieri, D., Becerra, S. P., & Steeg, P. S. (2012). Opposing effects of pigment epithelium-derived factor on breast cancer cell versus neuronal survival: Implication for brain metastasis and metastasis-induced brain damage. Cancer Research, 72(1), 144–153. https://doi.org/10.1158/0008-5472.CAN-11-1904

Hong, H., Zhou, T., Fang, S., Jia, M., Xu, Z., Dai, Z., Li, C., Li, S., Li, L., Zhang, T., Qi, W., Bardeesi, A. S. A., Yang, Z., Cai, W., Yang, X., & Gao, G. (2014). Pigment epithelium-derived factor (PEDF) inhibits breast cancer metastasis by down-regulating fibronectin. Breast Cancer Research and Treatment, 148(1), 61–72. https://doi.org/10.1007/s10549-014-3154-9

Jones, I. C., Carnagarin, R., Armstrong, J., Lin, D. P. L., Baxter-Holland, M., Elahy, M., & Dass, C. R. (2023). Pigment Epithelium-Derived Factor: Inhibition of Phosphorylation of Insulin Receptor (IR)/IR Substrate (IRS), Osteogeneration from Adipocytes, and Increased Levels Due to Doxorubicin Exposure. Pharmaceutics, 15(7). https://doi.org/10.3390/pharmaceutics15071960

Kawahara, K., Yoshida, T., Maruno, T., Oki, H., Ohkubo, T., Koide, T., & Kobayashi, Y. (2020). Spatiotemporal regulation of PEDF signaling by type I collagen remodeling. Proceedings of the National Academy of Sciences of the United States of America, 117(21). https://doi.org/10.1073/PNAS.2004034117

Kelland, L. R., & Abel, G. (1992). Comparative in vitro cytotoxicity of taxol and Taxotere against cisplatin-sensitive and -resistant human ovarian carcinoma cell lines. Cancer Chemotherapy and Pharmacology, 30(6), 444–450. https://doi.org/10.1007/BF00685595

Subramanian, P., Deshpande, M., Locatelli-Hoops, S., Moghaddam-Taaheri, S., Gutierrez, D., Fitzgerald, D. P., Guerrier, S., Rapp, M., Notario, V., & Becerra, S. P. (2012). Identification of pigment epithelium-derived factor protein forms with distinct activities on tumor cell lines. Journal of Biomedicine and Biotechnology, 2012. https://doi.org/10.1155/2012/425907

Tsuruhisa, S., Matsui, T., Koga, Y., Sotokawauchi, A., Yagi, M., & Yamagishi, S.-I. (2021). Pigment epithelium-derived factor inhibits advanced glycation end product-induced proliferation, VEGF and MMP-9 expression in breast cancer cells via interaction with laminin receptor. Oncology Letters, 22(2), 629. https://doi.org/10.3892/ol.2021.12890

Zhou, D., Xu, P., Zhang, M., Ye, G., & Zhang, L. (2017). Inhibitory effects of pigment epithelium-derived factor on epithelial-mesenchymal transition, migration and invasion of breast cancer. International Journal of Clinical and Experimental Pathology, 10(10), 10593–10602. http://www.ncbi.nlm.nih.gov/pubmed/31966401

Zhou, D., Zhang, M., Xu, P., Yu, Y., Ye, G., Zhang, L., & Wu, A. (2016). Expression of pigment epithelium-derived factor is associated with a good prognosis and is correlated with epithelial-mesenchymal transition-related genes in infiltrating ductal breast carcinoma. Oncology Letters, 11(1), 116–124. https://doi.org/10.3892/ol.2015.3880

Minor comments:

  1. I found that the manuscript is not well curated in terms of data representation, there are many errors in the text. In Fig. 1 (a,b) I would label the Y-axis as % total LT+ cells and cell types at the top or bottom of the graph. The graphs are not clear as there is inconsistency in the origin of the graph on the X-axis.

Thank you for the comment, all your suggestions have been incorporated.

  1. Fig. 1c, Y-axis percentage cannot be more than 100% percent.

Actually I think the graph was not clear, and it has been modified by adding a blue colour to the control line, and putting the word control in the same blue colour. We think it is now easier to interpret, i.e. the LT- cells grow more than the control in the same culture time, that is why they are above the blue line that marks the growth of the controls. We have also added in the figure caption the following clarification: "The growth of the control cells has been represented with a blue horizontal line".

  1. I would like to see the highest concentration of Docetaxel at which cell survival is reduced to 10% or so.

This is a very good suggestion, and in future work we will test it to find the concentration that will reduce resistant cells to below 10%. It will be taken into account for future articles on the subject. Thank you.

  1. In Fig. 1 all your suggestions have been incorporated.
  2. The quality of the graphs has been improved.
  3. Fig. 2. Cytoplasm is misspelled as Citoplasm, Cytoplamatic etc.

We have corrected the typo

  1. In Fig. 2 (H-K) in addition to ICC, I am curious to know the percentage of the population in FACS.

It is a very interesting question, and we are trying to do it. But it is not a simple experiment, because the population is still very much in the minority and we are often at the detection limit of the FACS instruments. This may be due, in our opinion, to the fact that the population of LT+ cells is not lost when the PEDF treatment is applied, but because they are slow dividing, there are not many progeny and the FACS technique has difficulties to show this population in a rigorous way. For this reason, we decided to detect the cells in ICC, although there are few, the difference is observed compared to the untreated cells. But it is undoubtedly an experiment that we will carry out in the future as soon as possible.

8.    We have improved the quality of the images as much as possible.

We have checked the English of the document with a native speaker.

Round 2

Reviewer 2 Report

Comments and Suggestions for Authors

The author Carmen Gil Gas and team has tried to address most of the concern raised. I understand there are some technical difficulty associated with the cell expansion and subsequent FACS analysis for different markers expressed on cell surface. 

I have no further suggestions to make. However, I would suggest to formally consult English language for final editing of the manuscript. 

Comments on the Quality of English Language

I have no further suggestions to make. However, I would suggest to formally consult English language for final editing of the manuscript. 

Author Response

Dear Reviewer

Thank you very much for taking the time to review this manuscript. We have revised the language with a Native speaker to improve the quality of English in the paper. We have uploaded the corrected manuscript and a certificate of proofreading. 

We hope that the article now meets the requirements to be published. Do not hesitate to contact us if we have to make any other correction.

 Best regards